# HierSpeech: Bridging the Gap between Text and Speech by Hierarchical Variational Inference using Self-supervised Representations for Speech Synthesis

**Sang-Hoon Lee**[1]     **Seung-Bin Kim**[2]     **Ji-Hyun Lee**[2]

**Eunwoo Song**[3]     **Min-Jae Hwang**[3]     **Seong-Whan Lee**[2*]

[1]Department of Brain and Cognitive Engineering, Korea University, Seoul, Korea
[2]Department of Artificial Intelligence, Korea University, Seoul, Korea
[3]NAVER Corp., Seongnam, Korea
[1,2]{sh_lee, sb-kim, jihyun-lee, sw.lee}@korea.ac.kr
[3]{eunwoo.song, min-jae.hwang}@navercorp.com

## Abstract

This paper presents HierSpeech, a high-quality end-to-end text-to-speech (TTS) system based on a hierarchical conditional variational autoencoder (VAE) utilizing self-supervised speech representations. Recently, single-stage TTS systems, which directly generate raw speech waveform from text, have been getting interest thanks to their ability in generating high-quality audio within a fully end-to-end training pipeline. However, there is still a room for improvement in the conventional TTS systems. Since it is challenging to infer both the linguistic and acoustic attributes from the text directly, missing the details of attributes, specifically linguistic information, is inevitable, which results in mispronunciation and over-smoothing problem in their synthetic speech. To address the aforementioned problem, we leverage self-supervised speech representations as additional linguistic representations to bridge an information gap between text and speech. Then, the hierarchical conditional VAE is adopted to connect these representations and to learn each attribute hierarchically by improving the linguistic capability in latent representations. Compared with the state-of-the-art TTS system, HierSpeech achieves +0.303 comparative mean opinion score, and reduces the phoneme error rate of synthesized speech from 9.16% to 5.78% on the VCTK dataset. Furthermore, we extend our model to HierSpeech-U, an untranscribed text-to-speech system. Specifically, HierSpeech-U can adapt to a novel speaker by utilizing self-supervised speech representations without text transcripts. The experimental results reveal that our method outperforms publicly available TTS models, and show the effectiveness of speaker adaptation with untranscribed speech.

## 1   Introduction

Text-to-speech (TTS) systems have undergone significant improvements in synthesizing high-quality speech from text sequence. Conventional TTS systems generally consist of two parts; an acoustic model and a vocoder. First, acoustic models (Wang et al., 2017; Shen et al., 2018) have shown the success of synthesizing acoustic features (e.g., Mel-spectrogram) as an intermediate feature from

---

*Corresponding author

text sequence, and the vocoder (Oord et al., 2016) converts the acoustic features into raw waveforms consecutively.

However, previous TTS models are subject to two limitations: 1) although speech consists of various attributes (e.g., pronunciation, rhythm, intonation, and timbre) (Qian et al., 2020; Choi et al., 2021), most previous models synthesize acoustic features from the text sequence at once (Ren et al., 2019), which exacerbates the one-to-many mapping problem; and 2) in the two-stage pipeline, each part of the TTS system should be trained independently, which results in the degradation of the audio quality (Ren et al., 2021a,b; Lee et al., 2021b).

Recently, single-stage end-to-end TTS models, which directly generate a raw waveform from text, successfully reduce these limitations of the two-stage pipeline. For instance, VITS (Kim et al., 2021) adopts variational inference augmented with the normalizing flow (Kim et al., 2020) and adversarial training (Kong et al., 2020) to improve the expressiveness of the model, which can learn rich representations from speech data and synthesize waveforms directly from the text. However, despite efforts to reduce the information gap between text and speech, these models are subject to speech mispronunciation and over-smoothing problems. In the process of synthesizing speech, they still generate all the acoustic attributes from text sequence at the same time. Therefore, missing the details of some attributes between text and speech, specifically linguistic information, is inevitable.

To bridge the information gap between text and speech, we adopt self-supervised speech representations as additional linguistic representations. Trained with large-scale speech dataset, these representations can learn useful information without using labeled data. Previous studies (Shah et al., 2021; Choi et al., 2021) also reveal that the representations from the pre-trained model contain rich information trained from a large-scale speech dataset. In particular, the representations from the middle layer of the pre-trained model contain rich linguistic information which has a characteristic of pronunciation. As a result, it has been successfully utilized for various speech tasks such as speech recognition (Baevski et al., 2020, 2021), voice conversion (Choi et al., 2021; Lee et al., 2021a), and speech resynthesis (Polyak et al., 2021). However, these useful representations have not yet received significant attention in TTS systems due to the difficulty to utilize in generative model.

In this paper, we present HierSpeech, which is a hierarchical conditional variational autoencoder using self-supervised speech representations for end-to-end TTS. We leverage self-supervised speech representations (Baevski et al., 2020) to enrich the linguistic information in latent representations, and to learn each attribute hierarchically from linguistic representations to acoustic representations. Although the self-supervised representations of text were previously studied for TTS (Jia et al., 2021), to the best of our knowledge, this is the first study that involves the investigation of self-supervised speech representations for single-stage end-to-end TTS. Based on the state-of-the-art TTS model (Kim et al., 2021), we demonstrate that self-supervised speech representations can reduce the information gap between text and speech by significantly improving the reconstruction quality. Compared with the state-of-the-art TTS model, HierSpeech achieves +0.303 comparative mean opinion score (CMOS) and reduces the phoneme error rate from 9.16 to 5.78 for the VCTK dataset.

Based on the pre-trained HierSpeech, we also present novel adaptive TTS frameworks. Specifically, we extend HierSpeech to HierSpeech-U, which can adapt the pre-trained model to synthesize the voice of novel speakers with untranscribed speech data. To extract the linguistic representations without text transcripts, HierSpeech-U can utilize the self-supervised speech representations, and learn the acoustic representations from untranscribed speech data. The results show that the synthetic quality of HierSpeech-U trained only with the speech data is comparable to that of HierSpeech trained with the text-speech pairs. The main contributions of this paper are as follows:

- We propose HierSpeech, which is a hierarchical conditional variational autoencoder using self-supervised representations that can improve the linguistic information in the latent representations, and learn attributes hierarchically. This significantly improves the reconstruction quality by bridging the gap between text and speech.

- We investigate self-supervised speech representations for the TTS system by thoroughly conducting more than 30,000 GPU hours of experiments. Audio samples are available at `https://sh-lee-prml.github.io/hierspeech-demo/`

- To utilize untranscribed speech data, we extend the model to HierSpeech-U which can adapt the TTS model without text transcripts. The results also reveal that the adaptation quality without text transcripts is comparable to that of the baseline model using text transcripts.

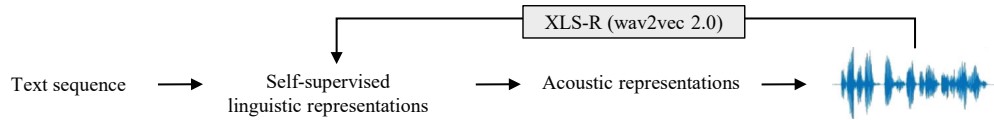

Figure 1: Hierarchical text-to-speech pipeline.

## 2 HierSpeech

In this paper, we propose the hierarchical conditional variational autoencoder using self-supervised representations for TTS. We use the self-supervised speech representations to improve the linguistic information in the latent representations, and to learn each representation hierarchically. Moreover, we extend HierSpeech to HierSpeech-U, which can adapt the model without text transcripts. The details of the speech representations, architecture, and untranscribed TTS method are described in the following subsections.

### 2.1 Speech representations

**Acoustic representations**   In TTS systems, Mel-spectrogram is widely used as an intermediate acoustic feature, which is converted from the waveform using the short-time Fourier transform (STFT). However, this acoustic feature consists of various attributes such as linguistic information (e.g., pronunciation) and style information (e.g., rhythm, intonation, and timbre). Hence, synthesizing this rich feature from only the text simultaneously exacerbates the one-to-many mapping problem, and it is difficult to extract the expressive linguistic information from the spectrogram. To mitigate this issue, we adopt additional linguistic features to map the text and acoustic features as follows.

**Linguistic representations**   To bridge the gap between text and speech, we use self-supervised speech representations for additional intermediate linguistic features as shown in Fig. 1. Previous studies reveal that the extracted features from the pre-trained model, such as wav2vec 2.0, contain rich linguistic information, and this can improve various tasks such as automatic speech recognition (ASR) and speech translation. (Shah et al., 2021; Choi et al., 2021) also shows the representations from the middle layer of wav2vec 2.0 contain a large proportion of linguistic information relevant to pronunciation. Specifically, we use the self-supervised speech representations from the 12th layer of the XLS-R (Babu et al., 2021) which is a pre-trained wav2vec 2.0 with a large-scale cross-lingual speech dataset. In addition, we also conduct various experiments to investigate these representations for the TTS system as detailed in Section 3.3 and Appendix C.

### 2.2 Hierarchical variational inference

To connect the two parts of TTS systems, the previous end-to-end TTS model, VITS (Kim et al., 2021) adopts a conditional variational autoencoder that maximizes the evidence lower bound (ELBO) over the intractable marginal log-likelihood of data $\log p_\theta(x|c)$:

$$\log p_\theta(x|c) \geq \mathbb{E}_{q_\phi(z|x)}\left[ \log p_\theta(x|z) - \log \frac{q_\phi(z|x)}{p_\theta(z|c)} \right] \tag{1}$$

where $p_\theta(z|c)$ is a prior distribution over latent variables $z$ given condition $c$, $p_\theta(x|z)$ is the likelihood function that generates data $x$ given latent variables $z$ as a decoder, and $q_\phi(z|x)$ is the approximate posterior. Subsequently, VITS uses the normalizing flow to improve the expressiveness of the prior distribution and adversarial training in the waveform domain. Based on VITS, HierSpeech uses a hierarchical conditional variational autoencoder to connect multi-level intermediate representations via disentangled latent variables of speech representations, and learns them in an end-to-end manner. Unlike recently proposed hierarchical VAE (Vahdat and Kautz, 2020; Lee et al., 2022a) which uses top-down path networks conditioning each other, we approximate them separately from different representations of speech. As shown in Fig 2, the acoustic posterior distribution and linguistic posterior distribution are encoded separately by acoustic encoder $\phi_a$ and linguistic encoder $\phi_l$ respectively. To disentangle each latent variable, we use the linear-scale spectrogram of the target speech $x_{spec}$ for the rich acoustic representations $z_a$, and the output of the 12th layer of XLS-R $x_{w2v}$ for the rich linguistic representations $z_l$. The optimization objective of HierSpeech can be expressed

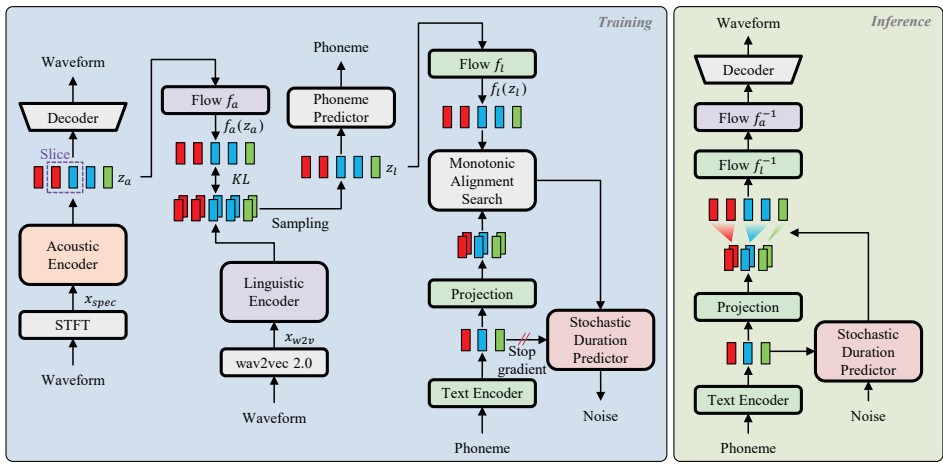

Figure 2: Overall framework of HierSpeech.

as follows:

$$\log p_\theta(x|c) \geq \mathbb{E}_{q_\phi(z|x)} \left[ \log p_{\theta_d}(x|z_a) - \log \frac{q_{\phi_a}(z_a|x_{spec})}{p_{\theta_a}(z_a|z_l)} - \log \frac{q_{\phi_l}(z_l|x_{w2v})}{p_{\theta_l}(z_l|c)} \right] \tag{2}$$

where $z = [z_a, z_l]$, $\theta = [\theta_d, \theta_a, \theta_l]$, $\phi = [\phi_a, \phi_l]$, $q_{\phi_a}(z_a|x_{spec})$ and $q_{\phi_l}(z_l|x_{w2v})$ are the approximate posterior for acoustic and linguistic representation, and $p_{\theta_l}(z_l|c)$ is a prior distribution over linguistic latent variables $z_l$ given condition c, $p_{\theta_a}(z_a|z_l)$ is a prior distribution over acoustic latent variables $z_a$, where $z_l$ is sampled from $q_{\phi_l}(z_l|x_{w2v})$, and $p_{\theta_d}(x|z_a)$ is the likelihood function that generates data $x$ given latent variables $z_a$ as a decoder $\theta_d$. For the reconstruction loss, we use Mel-reconstruction loss $\mathcal{L}_{rec}$ which minimizes the $l1$ distance of the Mel-spectrogram between the ground truth and reconstructed waveform using STFT and Mel-scale transformation.

**Acoustic encoder and waveform decoder**  The acoustic encoder $\phi_a$ is composed of non-casual WaveNet residual blocks which are layers of dilated convolutions with a gated activation unit and skip connection. Thereafter, the output is then fed to the projection layer to sample the acoustic representations $z_a$ from the mean and variance of the posterior distribution using the reparametrization trick. During training, the sliced $z_a$ is fed to a waveform decoder to reconstruct the raw audio $x$. We use HiFi-GAN generator $G$ (Kong et al., 2020) as waveform decoder $\theta_d$ which consists of a stack of transposed convolution and multi-receptive field fusion module. For adversarial feedback, we also use the multi-period discriminator $D$ to capture the different periodic features of the waveform.

$$\mathcal{L}_{adv}(D) = \mathbb{E}_{(x,z_a)} \left[ (D(x) - 1)^2 + D(G(z_a))^2 \right], \tag{3}$$

$$\mathcal{L}_{adv}(\phi_a, \theta_d) = \mathbb{E}_{(z_a)} \left[ (D(G(z_a)) - 1)^2 \right] \tag{4}$$

where x is the ground truth waveform. To ensure stable training, we use the additional feature matching loss $\mathcal{L}_{fm}$ which minimizes the $l1$ distance of each discriminator's features between the ground truth and reconstructed waveform.

**Linguistic encoder and phoneme predictor**  The linguistic encoder $\phi_l$ has the same structure as the acoustic encoder. However, we use self-supervised speech representations $x_{w2v}$ as the input of the linguistic encoder which is extracted from the pre-trained XLS-R, and the linguistic representations $z_l$ is extracted. To enforce linguistic characteristics, $z_l$ is fed to the auxiliary phoneme predictor. We minimize the connectionist temporal classification (CTC) loss $\mathcal{L}_{ctc}$ to optimize the linguistic encoder and phoneme predictor. However, to remove the additional projection layer, the projected mean and variance from the linguistic encoder are also directly used as the acoustic prior distribution with weight-sharing between $\theta_a$ and $\phi_l$. To maintain hierarchy, the representation $z_a$ is transformed by normalizing flow. Hence, the KL divergence between acoustic prior and posterior is minimized as:

$$\mathcal{L}_{kl1} = \log q_{\phi_a}(z_a|x_{spec}) - \log p_{\theta_a}(z_a|x_{w2v}) \tag{5}$$

Because the acoustic prior distribution is obtained from the linguistic information, to bridge the gap between distributions, we use the normalizing flow to disentangle the information from the acoustic posterior and increase the expressiveness of the acoustic prior distribution.

$$p_{\theta_a}(z_a|x_{w2v}) = \mathcal{N}(f_a(z_a); \mu_{\theta_a}(x_{w2v}), \sigma_{\theta_a}(x_{w2v}))|\det(\partial f_a(z_a)/\partial z_a)|,$$
$$z_a \sim q_{\phi_a}(z_a|x_{spec}) = \mathcal{N}(z_a; \mu_{\phi_a}(x_{spec}), \sigma_{\phi_a}(x_{spec})) \tag{6}$$

**Text encoder**   The text encoder $\theta_l$ consists of a stacked feed-forward Transformer network using a relative positional representation. The phoneme sequence $c_{text}$ is fed to the text encoder, and the projection layer is used to produce the mean and variance for linguistic prior distribution. To align the text with the linguistic representations of speech, we use monotonic alignment search (MAS) of (Kim et al., 2020) which searches the alignment $A$ satisfied with maximizing the likelihood of data:

$$\mathcal{L}_{kl2} = \log q_{\phi_l}(z_l|x_{w2v}) - \log p_{\theta_l}(z_l|c_{text}, A) \tag{7}$$

We also use the normalizing flow to increase the expressiveness of the linguistic prior distribution.

$$p_{\theta_l}(z_l|c_{text}, A) = \mathcal{N}(f_l(z_l); \mu_{\theta_l}(c_{text}, A), \sigma_{\theta_l}(c_{text}, A))|\det(\partial f_l(z_l)/\partial z_l)|,$$
$$z_l \sim q_{\phi_l}(z_l|x_{w2v}) = \mathcal{N}(z_l; \mu_{\phi_l}(x_{w2v}), \sigma_{\phi_l}(x_{w2v})) \tag{8}$$

To sample the duration given phonemes, we adopt a flow-based stochastic duration predictor of (Kim et al., 2021) trained with maximum likelihood estimation. We use the negative variational lower bound of it as duration loss $\mathcal{L}_{dur}$. For the multi-speaker settings, we add the global speaker embedding to the residual block of the acoustic/linguistic encoder, residual block in normalizing flow, stochastic duration predictor, and decoder.

**Total loss**   The final objectives for HierSpeech can be expressed as follows:

$$\mathcal{L}_{total} = \mathcal{L}_{kl1} + \lambda_{kl2}\mathcal{L}_{kl2} + \lambda_{rec}\mathcal{L}_{rec} + \lambda_{ctc}\mathcal{L}_{ctc} + \lambda_{dur}\mathcal{L}_{dur} + \lambda_{adv}\mathcal{L}_{adv}(\phi_a, \theta_d) + \lambda_{fm}\mathcal{L}_{fm} \tag{9}$$

### 2.3   Untranscribed text-to-speech

For the untranscribed TTS model (HierSpeech-U), we train the model using a style encoder (Min et al., 2021) which extracts style embedding from speech as global conditioning (Jia et al., 2018). We use a linear-scale spectrogram as the input of the style encoder. After pre-training with the multi-speaker dataset, we adapt the model to a novel speaker without text transcripts. Through self-supervised speech representations, the pre-trained linguistic encoder is able to extract rich linguistic representations from speech without text transcripts. Hence, HierSpeech-U synthesizes speech with the style of a novel speaker by fine-tuning the acoustic encoder, the normalizing flow blocks of the acoustic prior, and decoder with only speech data.

## 3   Experiment and Result

### 3.1   Experimental setup

**Datasets**   We train the models using the VCTK and LibriTTS datasets. The VCTK dataset contains 46 hours of audio for 108 speakers (Veaux et al., 2017). We use the *train-clean* subsets of the LibriTTS dataset (Zen et al., 2019) which contains 110 hours of audio for 1,151 speakers. For both datasets, we downsample the audio at 22,050Hz. Following the (Lee et al., 2021a), we use only non-parallel data for the training dataset which consists of different utterances for each speaker, and we use the parallel data for the test dataset which consists of 25 same utterances in the VCTK dataset. For the LibriTTS dataset, we randomly select two samples from each speaker for the test dataset. For speaker adaptation, we randomly select 98 speakers as the base speakers and 10 speakers (5 males and 5 females) as the novel speakers. To evaluate the speaker adaptation performance for the number of speakers, we also train the model using 1,249 speakers from the VCTK and LibriTTS datasets.

**Preprocessing**   For the input of the acoustic and speaker encoder, we use linear-scale spectrogram with 513 bins which is transformed from the audio. For reconstruction loss, we use a Mel-spectrogram with 80 bins. For the input of the linguistic encoder, we use the output from the middle layer of the XLS-R (0.3B) (Babu et al., 2021) which is pre-trained wav2vec 2.0 with a large-scale cross-lingual

speech dataset.[2] To extract the self-supervised speech representations, we use the downsampled audio at 16,000Hz as an input to the XLS-R, and the extracted representations are upsampled to map the spectrogram by interpolation.[3] For the input of the text encoder, we use the text sequences converted to International Phonetic Alphabet (IPA) sequences using the open-source Phonemizer.[4] Following (Kim et al., 2021), we intersperse the phoneme sequences with a blank token. However, we use phoneme sequences without a blank token for the target sequence of the phoneme predictor.[5]

**Training**    We train HierSpeech using the AdamW optimizer (Loshchilov and Hutter, 2019) with $\beta_1 = 0.8$, $\beta_2 = 0.99$, and weight decay $\lambda = 0.01$, and apply the learning rate schedule by the decay of $0.999^{1/8}$ at an initial learning rate of $2 \times 10^{-4}$ for the generator and discriminator. We train HierSpeech with a batch size of 256 for 600k steps using mixed precision training on four NVIDIA A100 GPUs. For speaker adaptation, we fine-tune HierSpeech/HierSpeech-U with few samples of novel speakers for only 2000 steps. For the ablation study, we train the model with a batch size of 64 for 300k steps.

## 3.2   Evaluation metrics

**Subjective metrics**    We conduct two mean opinion score (MOS) tests for naturalness and similarity. For the naturalness MOS (nMOS), each sample from the target and synthesized speech are rated by at minimum of 20 listeners on a scale of 1-5. For the similarity MOS (sMOS), the synthesized and target speech are presented to a minimum of 20 listeners on a scale of 1-4. The nMOS and sMOS are reported with 95% confidence intervals.

**Objective metrics**    For the naturalness measurement, we calculate the phoneme error rate (PER) and word error rate (WER) of the synthesized speech. We used the fined-tuned wav2vec 2.0 (Baevski et al., 2020) without a language model for automatic speech recognition. For the similarity measurement, we conduct three metrics: the equal error rate (EER) of the automatic speaker verification (ASV), the Mel-cepstral distortion (MCD), and the F0 root mean square error ($\text{RMSE}_{f0}$). We used a pre-trained ASV model (Chung et al., 2020). The ASV model is trained using a large-scale speech dataset, VoxCeleb2 (Chung et al., 2018) via online data augmentation (Heo et al., 2020). We calculate the EER where both acceptance and rejection rates are equal for the sample pairs from the synthesized and target speech ($2,700 \times 108 = 291,600$). We compute the MCD and $\text{RMSE}_{f0}$ by applying dynamic time warping (DTW) between the synthesized speech and target speech. We conducted a duration prediction performance evaluation using the average absolute differences of the utterance duration (DDUR) (Zhang et al., 2019). To compare the inference speeds, we calculate the Speed which is the synthetic waveform frame per second, and the synthesis speed over real-time (Real-time).

## 3.3   Analysis of self-supervised representations

Previous researches (Fan et al., 2020; Choi et al., 2021) reveal that the representations from the front layer of wav2vec 2.0 are clustered by each speaker. Hence, (Choi et al., 2021) uses the 1st layer of pre-trained wav2vec 2.0 as an input to the speaker encoder. As an input of the speaker encoder, we compared three different features; linear-scale spectrogram (linear spectrogram), the representations from the 1st layer of XLS-R and 12th layer of XLS-R. Table 1 shows that the linear spectrogram has better transfer performance than the

Table 1: Results of different input features for the speaker encoder.

| Input feature | PER | WER | EER |
|---|---|---|---|
| Linear spectrogram | 5.45 | 20.13 | 2.77 |
| 1st layer of XLS-R | 5.67 | 20.36 | 4.62 |
| 12th layer of XLS-R | 6.29 | 21.50 | 10.23 |

representations of XLS-R. Although the speaker embedding extracted from the 1st layer of the XLS-R has better performance than one of the 12th layer, the results show that the linear spectrogram contains a significant amount of high-resolution information for speaker characteristics. In this respect, we use linear spectrogram as the input for the speaker encoder, and we can conclude that there is a little speaker information to transfer speaking style in the representations of the middle layer of XLS-R.

---

[2]Specifically, we use the 12th layer of the XLS-R. The results are detailed in Section 3.3 and Appendix C.

[3]We simply interpolate with respect to the time-domain. The ASR model uses the audio with a sampling rate of 16,000 Hz. However, the TTS model use higher resolution audio for high-quality audio synthesis.

[4]`https://github.com/bootphon/phonemizer`

[5]We observe that the blank token degraded the classification performance under the experimental conditions.

To evaluate the speech disentanglement performance with respect to the layer of XLS-R, we conduct frame-level speaker classification on the representations from a specific layer of the XLS-R and the linguistic latent variables $z_l$ respectively. Table 2 shows that the representations from each layer contain some speaker information to be classified. However, the speaker information is reduced through the linguistic encoder, and the $z_l$ from the 12th layer of XLS-R has the lowest accuracy. Hence, we use representations from the 12th layer for

Table 2: Speaker classification accuracy on linguistic representations from the different layer of XLS-R.

| layer | Accuracy (w2v) | Accuracy ($z_l$) |
|---|---|---|
| 1st layer | 79.12% | 15.09% |
| 12th layer | 59.46% | 8.95% |
| 24th layer | 48.73% | 11.35% |

the linguistic encoder. It should be noted that all representations, with the exception of the 23th layer, improve the TTS performance of HierSpeech when compared with that of VITS, which means that all representations contain rich information trained with large-scale speech dataset.

## 3.4 Evaluation on TTS

Table 3: The TTS evaluation results on the VCTK dataset.

| Method | nMOS | sMOS | PER | WER | EER | MCD | RMSE$_{f0}$ | DDUR | Speed (kHz) | Real-time |
|---|---|---|---|---|---|---|---|---|---|---|
| GT | 4.06±0.02 | 3.34±0.03 | 5.64 | 18.94 | 4.03 | - | - | - | - | - |
| GT (HiFi-GAN) | 4.03±0.02 | 3.30±0.03 | 5.94 | 19.52 | 5.04 | 1.25 | 28.32 | - | 6,484.09 | ×294.06 |
| Tacotron2 | 3.76±0.02 | 3.16±0.03 | 11.73 | 22.48 | 9.11 | 4.18 | 35.30 | 0.49 | 263.94 | ×11.97 |
| Glow-TTS | 3.95±0.02 | 3.09±0.03 | 11.77 | 26.40 | 5.33 | 4.31 | 32.98 | 0.38 | 1,410.75 | ×63.97 |
| PortaSpeech | 3.97±0.02 | 3.15±0.03 | 11.35 | 25.46 | 5.48 | 4.34 | 32.89 | 0.43 | 1,163.21 | ×52.75 |
| VITS | 4.02±0.02 | 3.19±0.03 | 9.16 | 25.54 | 3.83 | 4.27 | 32.93 | 0.37 | **1,610.77** | **×72.83** |
| HierSpeech (Ours) | **4.04±0.02** | **3.22±0.03** | **5.78** | **19.55** | **3.74** | **4.05** | **32.15** | **0.33** | 1,459.95 | ×66.21 |

Table 4: The speaker transfer evaluation results on the LibriTTS dataset.

| Method | nMOS | sMOS | PER | WER | EER | MCD | RMSE$_{f0}$ | DDUR | Speed (kHz) | Real-time |
|---|---|---|---|---|---|---|---|---|---|---|
| GT | 4.04±0.03 | 3.40±0.03 | 7.01 | 18.28 | 4.45 | - | - | - | - | - |
| VITS | 3.96±0.03 | 3.26±0.03 | 13.62 | 29.83 | 5.00 | **4.37** | 34.18 | 1.09 | **1,781.40** | **×80.78** |
| HierSpeech (Ours) | **3.98±0.03** | 3.26±0.03 | **7.47** | **20.34** | 5.00 | 4.42 | **32.95** | **0.72** | 1,678.79 | ×76.13 |

Table 3 shows that our model outperforms the other models with respect to the nMOS and sMOS for both datasets. In terms of the ASR evaluation, our model shows a lower PER and WER than the other models by synthesizing speech with more accurate pronunciation. In terms of EER, all models have similar performance, which indicates the target speaker embedding (ID) is useful supervision for multi-speaker TTS. For the speaker transfer, we evaluate each model using the same speaker encoder in Table 4 and subsection 3.5. In terms of the MCD and RMSE$_{f0}$, our model has the lowest error distance. Although VITS has better performance in inference speed, our model has a faster inference speed than two-stage end-to-end TTS models. We also compare the models using speaker encoder, which are trained with VCTK and LibriTTS dataset in Table 4. Our model outperforms VITS in terms of nMOS and ASR evaluation. However, our model has a slightly lower transfer performance in MCD. We found that transferring the speaker from the long sentence or noisy audio results in low performance of ASR for VITS and low speaker transfer quality for HierSpeech. We also conduct the comparative mean opinion score (CMOS) tests between the models trained with each dataset; VCTK and LibriTTS. Table 5 also shows that our model has a better performance than VITS in CMOS evaluation with $t$-test $p$-values. Also, our model achieves -0.096 CMOS compared to ground truth (GT) audio on the VCTK dataset. However, Table 5b shows that our model has -0.505 CMOS compared to GT, and it means that there is a room for improvement in TTS system by improving the expressiveness and robustness of the model.

Table 5: CMOS comparison. Positive score indicates that HierSpeech is rated better than the baseline.

(a) The CMOS results on the VCTK dataset.

| Method | CMOS | p-value |
|---|---|---|
| HierSpeech (Ours) | 0 | - |
| GT | -0.096 | 0.003 |
| VITS | +0.303 | $<10^{-24}$ |

(b) The CMOS results on the LibriTTS dataset.

| Method | CMOS | p-value |
|---|---|---|
| HierSpeech (Ours) | 0 | - |
| GT | -0.505 | $<10^{-104}$ |
| VITS | +0.297 | $<10^{-25}$ |

Table 6: Results for untranscribed text-to-speech. We compare few-shot speaker adaptation performance of HierSpeech-U with that of HierSpeech. Both models use the pre-trained HierSpeech which is trained using VCTK and LibriTTS datasets. We used 10 unseen speakers of VCTK dataset as novel speakers, and fine-tuned each model with 20 samples from each speaker.

| Method | Transcript | nMOS | sMOS | PER | WER | EER | MCD | RMSE$_{f0}$ | DDUR |
|---|---|---|---|---|---|---|---|---|---|
| GT | - | 4.13±0.10 | 3.38±0.10 | 4.26 | 16.69 | 4.14 | - | - | - |
| HierSpeech | ✓ | 4.09±0.10 | 3.18±0.11 | 4.40 | 16.95 | 6.40 | 3.96 | 29.56 | 0.28 |
| HierSpeech-U | ✗ | 4.08±0.09 | 3.15±0.12 | 3.71 | 15.85 | 6.40 | 4.09 | 30.64 | 0.36 |

## 3.5 Untranscribed text-to-speech

To evaluate the untranscribed text-to-speech performance of HierSpeech-U, we compare the few-shot speaker adaptation performance of HierSpeech-U with one of HierSpeech which is fine-tuned by unseen speakers using few audio-text pairs (Arik et al., 2018). Both models use the pre-trained HierSpeech which is trained using VCTK and LibriTTS datasets. For the speaker adaptation of HierSpeech, we fine-tune the entire model. Although HierSpeech-U is fine-tuned without text transcripts, Table 6 shows that HierSpeech-U has comparable performance to HierSpeech in terms of nMOS and sMOS. Moreover, HierSpeech-U achieves a lower PER and WER than that of HierSpeech, which is slightly increased compared to PER 3.58 and WER 15.50 for the pre-trained model. Although there is a limitation in adapting duration, HierSpeech-U is able to adapt the speaker in terms of voice by utilizing the linguistic representations from the self-supervised speech representations without text transcripts. Note that we fail to fine-tune VITS without text transcripts in that the PER for VITS increases from 7.47 to 12.27. Furthermore, we evaluate the adaptation quality with different numbers of adaptation samples (1, 5, 10, and 20) in Figure 3. We also investigate the effectiveness for different numbers of pre-trained speakers (98 from VCTK and 1,151 from LibriTTS) and the evaluation results are described in Appendix C.

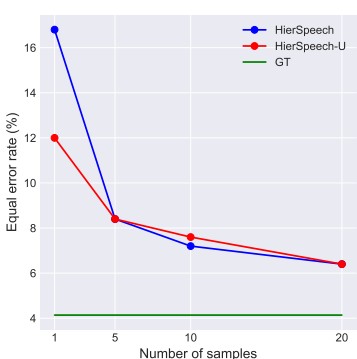

Figure 3: The EER results for different numbers of adaptation samples.

## 3.6 Ablation study and hyperparameter search

To compare the effectiveness for the number of normalizing flow, we train VITS which has the same number of flow blocks ($4 \rightarrow 8$) with HierSpeech. However, we observe that increasing the number of flow blocks does not always indicate an improvement in model performance, resulting in the degradation of the pronunciation and audio quality as shown in Table 7. Adding a phoneme predictor (PP) to the posterior encoder of VITS improves the PER and WER by guiding the alignment search. Removing the acoustic encoder (AE) and synthesizing waveform directly from the linguistic representations from the 12th layer of XLS-R degrade the performance, and it also has higher Mel-spectrogram reconstruction error.

Table 7: Results of the ablation study.

| Method | PER | WER | EER | MCD |
|---|---|---|---|---|
| VITS | 12.24 | 30.62 | 3.85 | 4.36 |
| w flow 8 | 13.42 | 32.77 | 4.00 | 4.36 |
| w PP | 7.60 | 22.98 | 3.74 | 4.17 |
| HierSpeech | **6.25** | **20.89** | **3.48** | **4.15** |
| w.o AE | 8.00 | 23.84 | 3.63 | 4.25 |
| w.o AE and PP | 16.70 | 37.43 | 3.89 | 4.39 |

We conduct a hyperparameter search and ablation study of the phoneme predictor in Appendix C. The proper value of $\lambda_{ctc} = 45$ increases the overall performance in all the objective metrics. In contrast, when using an excessively large $\lambda_{ctc}$, the KL divergence between the acoustic and linguistic distribution increases, which decreases the performance of audio quality with noisy sound. For better generalization, we attempted to combine label smoothing with CTC loss (Kim et al., 2018) and conduct experiments on data augmentation for the effective speech disentanglement of linguistic information following (Choi et al., 2021). However, label smoothing and data augmentation lead to the problem for KL divergence optimization, thus resulting in the degradation of the synthesis quality.

### 3.7 Evaluation on VC

To evaluate speech disentanglement, we compared the voice conversion (VC) task of our model with three models; AutoVC (Qian et al., 2019), VoiceMixer (Lee et al., 2021a), and VITS (Kim et al., 2021). For a fair comparison, we train all models using the same dataset. However, note that AutoVC and VoiceMixer do not use the phoneme information. Following (Lee et al., 2021a), we randomly select 20 speakers. Thereafter, a single audio sample is selected from each speaker, and then all the possible pairs of samples ($20 \times 20 = 400$) are produced for evaluation. Table 8 shows that our model outperforms the other models in nMOS and ASR evaluation metrics. Although VITS has similar performance in the EER evaluation, our model has better performance in ASR evaluation, which means that our model can disentangle content and style information with a small loss of content information. We also observe that using the representations from the middle layer of XLS-R improves the PER performance when compared with that of the previous layer, and the auxiliary phoneme predictor also helps retain the linguistic information.

Table 8: The VC evaluation results on the VCTK.

| Model | nMOS | sMOS | PER | WER | EER |
|---|---|---|---|---|---|
| GT | 4.19±0.03 | 3.37±0.04 | 5.20 | 14.05 | 0.98 |
| AutoVC | 3.74±0.05 | 3.04±0.05 | 59.42 | 89.01 | 12.50 |
| VoiceMixer | 4.00±0.05 | 3.11±0.05 | 12.67 | 31.80 | 9.35 |
| VITS | 4.06±0.04 | 3.16±0.05 | 6.46 | 18.73 | **2.75** |
| HierSpeech | **4.08±0.04** | 3.16±0.05 | **5.51** | **16.25** | **2.75** |

## 4 Broader Impact

Recently, self-supervised speech representations have been utilized in TTS tasks (Siuzdak et al., 2022; Kim et al., 2022; Du et al., 2022) and we see that our hierarchical speech synthesis structure using self-supervised speech representations can be also utilized for various tasks. First, the application of a low-resource language can utilize self-supervised speech representations to improve the synthesis quality. Second, cross-lingual speaker adaptation for dubbing can be used in the film industry. However, as recent speech synthesis systems such as TTS and VC can generate the audio with realistic sound, there is an increased potential risk of harm, malicious use, and ethical issues. Specifically, these systems could be misused in various manners, such as fake news generation, voice spoofing, and unauthorized use of web crawl speech data. To mitigate these issues, fake audio detection is studied (Singhal et al., 2019; Tak et al., 2021). Moreover, we provide a mitigation strategy for the proposed system by releasing the fake audio detection model with an ensemble of discriminators that can distinguish real and fake audio based on results of multiple discriminators. Although it is difficult to differentiate in human evaluation score, the model achieves a 16.59% EER for fake audio detection on 1,852 test samples of the ground-truth and synthesized speech from our model.

## 5 Conclusion

We presented an end-to-end TTS model, HierSpeech, which can learn and synthesize speech from text through hierarchical intermediate representations in an end-to-end manner. By bridging the gap between text and speech through self-supervised speech representations, the proposed model significantly improved the reconstruction quality. We successfully demonstrated that our hierarchical conditional variational autoencoder can improve linguistic capability in latent representations, and learn each attribute hierarchically using self-supervised speech representations. We thoroughly conducted more than 30,000 GPU hours of experiments on self-supervised representations for the TTS system, and we hope that these results can serve as a basis for future speech research. Furthermore, we also demonstrate the effectiveness of a novel speaker adaptation framework without text transcripts. We see our hierarchical structure extending to cross-lingual TTS systems or other low-resource TTS systems. Also, we will try to improve the expressive and robustness of model by modeling prosody (Im et al., 2022) and noise from speech (Saeki et al., 2022). However, the single-stage end-to-end TTS model is limited in terms of computational complexity in that the training process requires 20 days using four A100 GPUs. Hence, in future works, an attempt will be made to decrease the computational cost without quality degradation by adopting iDWT (Lee et al., 2022b) in the decoder and simplifying the discriminator to facilitate more rapid training (Andreev et al., 2022), and replace the decoder with an diffusion-based neural vocoder (Koizumi et al., 2022b,a). In addition, we discussed the potential positive and negative impact of our model in Section 4. To mitigate the negative impact, we will release fake audio detector that can distinguish between real and fake audio.

## Acknowledgements

This work was supported by Institute of Information & communications Technology Planning & Evaluation (IITP) grant funded by the Korea government(MSIT) (No. 2019-0-00079, Artificial Intelligence Graduate School Program(Korea University), No. 2019-0-01371, Development of Brain-inspired AI with Human-like Intelligence, No. 2021-0-02068, Artificial Intelligence Innovation Hub, and No. 2022-0-00984, Development of Artificial Intelligence Technology for Personalized Plug-and-Play Explanation and Verification of Explanation) and Clova Voice, NAVER Corp., Seongnam, Korea.

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
