# A   Implementation Details

Table 9: Hyperparameters of HierSpeech.

|  | Hyperparameter | HierSpeech |
|---|---|---|
| Text Encoder | Phoneme Embedding | 192 |
|  | Layers | 6 |
|  | Hidden Size | 192 |
|  | Conv1D Kernel | 3 |
|  | Conv1D Filter Size | 768 |
| Linguistic/Acoustic Encoder | Input Feature | 12th layer of XLS-R/linear spectrogram |
|  | Input Size | 1024/513 |
|  | WaveNet Layers | 16 |
|  | WaveNet Channel Size | 192 |
|  | Conv1D Kernel | 5 |
| Flow ($f_a$ and $f_l$) | Affine Coupling Layers | 4 |
|  | Affine Coupling Dilation | 1 |
|  | Affine Coupling WaveNet Layers | 4 |
|  | Affine Coupling Kernel Size | 5 |
|  | Affine Coupling Filter Size | 192 |
| Decoder | Initial Hidden Size | 512 |
|  | MRF Kernel Size | [3,7,11] |
|  | MRF Dilation Size | [[1,3,5],[1,3,5],[1,3,5]] |
|  | Upsampling Rate | [8,8,2,2] |
|  | Upsampling Transposed Conv1D Kernel Size | [16,16,4,4] |
| Loss | $\lambda_{kl2}/ \lambda_{rec}/\lambda_{ctc}/\lambda_{dur}/\lambda_{adv}/ \lambda_{fm}$ | 1/45/45/1/1/2 |

The details of hyperparameter are described in Table 9. Moreover, we describe the objective function with the respect to the trained parameters in details as follows:

$$\mathcal{L}_{total} = \mathcal{L}_{kl1}(\phi_a, \theta_a, \theta_{f_a}) + \lambda_{kl2}\mathcal{L}_{kl2}(\phi_l, \theta_l, \theta_{f_l}) + \lambda_{rec}\mathcal{L}_{rec}(\phi_a, \theta_d) + \lambda_{ctc}\mathcal{L}_{ctc}(\phi_l, \theta_{pp})$$
$$+ \lambda_{dur}\mathcal{L}_{dur}(\theta_{dp}) + \lambda_{adv}\mathcal{L}_{adv}(\phi_a, \theta_d) + \lambda_{fm}\mathcal{L}_{fm}(\phi_a, \theta_d) \tag{10}$$

where $\theta_{f_a}$ is the parameter of normalizing flow network in acoustic prior encoder, $\theta_{f_l}$ is the parameter of normalizing flow network in linguistic prior encoder, and $\theta_{pp}$ and $\theta_{dp}$ is the phoneme predictor and duration predictor.

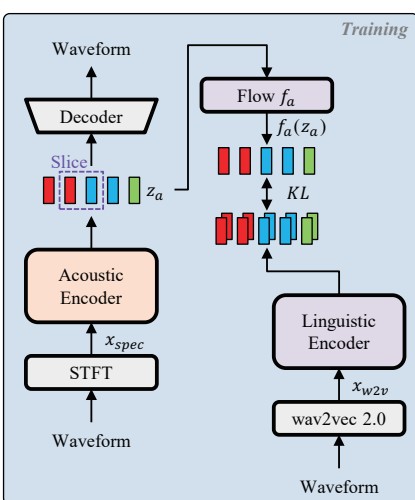

Figure 4: The training procedure of untranscribed text-to-speech.

For untranscribed text-to-speech as shown in Figure 4, we fine-tune the model without linguistic prior encoder, phoneme predictor, and duration predictor as follows objective:

$$\mathcal{L}_{finetuning} = \mathcal{L}_{kl1}(\phi_a, \theta_a, \theta_{f_a}) + \lambda_{rec}\mathcal{L}_{rec}(\phi_a, \theta_d) + \lambda_{adv}\mathcal{L}_{adv}(\phi_a, \theta_d) + \lambda_{fm}\mathcal{L}_{fm}(\phi_a, \theta_d) \tag{11}$$

# B   Voice Conversion

By disentangling speech representations into linguistic and acoustic representations, we assume that the linguistic representations contain a small amount of speaker information. To evaluate the speech disentanglement, we conduct the voice conversion task using the linguistic representations. In our model, we can extract the linguistic representations by four different methods.

First, we convert the speech $x$ into a linear spectrogram $x_{spec}$, and we predict the linguistic representations $\hat{z}_l$ through the acoustic encoder $q_{\phi_a}$ and the normalizing flow in the acoustic prior encoder $f_a$ given the source speaker information $s$ and linear spectrogram $x_{spec}$. Using the predicted linguistic representations $\hat{z}_l$, we generate a converted speech $x_t$ with a voice of target speaker information $s_t$ through the inverse transformation of the normalizing flow $f_a^{-1}$ and decoder $G$ as follows:

$$
\begin{aligned}
z_a &\sim q_{\phi_a}(z_a | x_{spec}, s) \\
\hat{z}_l &= f_a(z_a | s) \\
x_t &= G(f_a^{-1}(\hat{z}_l | s_t) | s_t)
\end{aligned}
\tag{12}
$$

Second, we convert the predicted linguistic representations $\hat{z}_l$ into a speaker-independent representations $e$ through the normalizing flow in the linguistic prior encoder $f_l$ given the source speaker information $s$. Using $e$ and the target speaker information $s_t$, we generate a converted speech $s_t$ through inverse transformation of the normalizing flow $f_l^{-1}$ and $f_a^{-1}$, and decoder $G$ as follows:

$$
\begin{aligned}
z_a &\sim q_{\phi_a}(z_a | x_{spec}, s) \\
\hat{z}_l &= f_a(z_a | s) \\
e &= f_l(\hat{z}_l | s) \\
x_t &= G(f_a^{-1}(f_l^{-1}(e | s_t) | s_t) | s_t)
\end{aligned}
\tag{13}
$$

Third, we extract the linguistic representations $z_l$ directly through the linguistic encoder $q_{\phi_l}$ given the source speaker information $s$ and the representations from the XLS-R $x_{w2v}$. Then, we generate a converted speech $x_t$ with a voice of target speaker information $s_t$ through the inverse transformation of the normalizing flow $f_a^{-1}$ and decoder $G$ as follows:

$$
\begin{aligned}
z_l &\sim q_{\phi_l}(z_l | x_{w2v}, s) \\
x_t &= G(f_a^{-1}(z_l | s_t) | s_t)
\end{aligned}
\tag{14}
$$

Finally, we convert the extracted linguistic representations $z_l$ into a speaker-independent representations $e$ through the normalizing flow in the linguistic prior encoder $f_l$ given the source speaker information $s$. Using $e$ and the target speaker information $s_t$, we generate a converted speech $s_t$ through inverse transformation of the normalizing flow $f_l^{-1}$ and $f_a^{-1}$ consecutively, and decoder $G$. The objectives for fine-tuning are represented as:

$$
\begin{aligned}
z_l &\sim q_{\phi_l}(z_l | x_{w2v}, s) \\
e &= f_l(z_l | s) \\
x_t &= G(f_a^{-1}(f_l^{-1}(e | s_t) | s_t) | s_t)
\end{aligned}
\tag{15}
$$

We conduct the ASR evaluation and ASV evaluation to compare the above methods. Table 10 shows that the first method with Eq. 12 has better performance than others, and this means that $x_{spec}$ contains more information to reconstruct the audio than $x_{w2v}$ alongside the ablation study of Table 7. The EER results also show that all method can convert the voice of speech, indicating that speech is disentangled through linguistic and acoustic encoder.

Table 10: Comparison for the different methods of voice conversion.

| Model | CER | PER | WER | EER |
|---|---|---|---|---|
| HierSpeech (Eq. 12) | 6.37 | 5.65 | 18.74 | 2.77 |
| HierSpeech (Eq. 13) | 8.42 | 7.96 | 20.34 | 2.92 |
| HierSpeech (Eq. 14) | 18.38 | 16.56 | 36.67 | 2.92 |
| HierSpeech (Eq. 15) | 20.15 | 18.89 | 40.09 | 3.72 |

# C Experiments

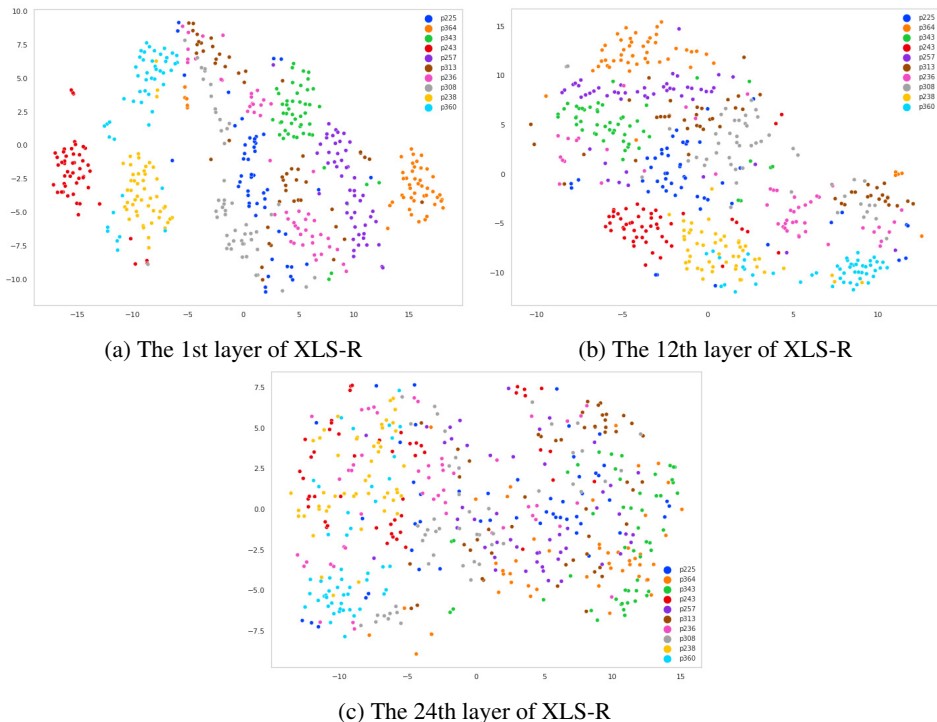

(a) The 1st layer of XLS-R

(b) The 12th layer of XLS-R

(c) The 24th layer of XLS-R

Figure 5: The t-SNE visualization for the representations from the different layers of XLS-R.

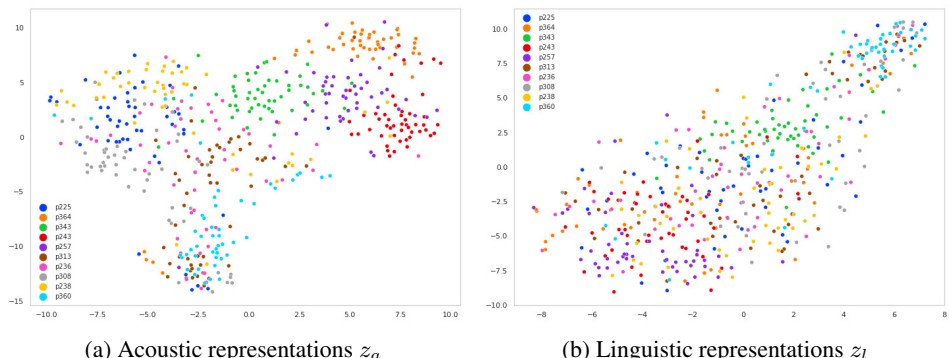

(a) Acoustic representations $z_a$

(b) Linguistic representations $z_l$

Figure 6: The t-SNE visualization for the acoustic/linguistic representations.

**t-SNE visualization**  In Figure 5, we present the t-SNE visualization for the representations from the different layers of XLS-R. Following (Choi et al., 2021), we average each representation from the 1st, 12th, and 24th layer of XLS-R on the time-axis for 50 utterances of 10 speakers. Similar to the previous analysis of XLSR-53 (Choi et al., 2021), the representations from the 1st layer of XLS-R are already clustered by each speaker while it is hard to distinguish the representations of the latter layer by each speaker. In Figure 6, we also present the t-SNE visualization for the acoustic representations $z_a$ and linguistic representations $z_l$. While acoustic representations are clustered by each speaker, it is difficult to differentiate the linguistic representations by speakers. This means that the linguistic encoder is able to extract the linguistic information from speech, and this allow the model to learn each representation hierarchically. It is worth noting that using different features from the same speech can simply extract the disentangled representations of speech in our model.

Table 11: Results for speaker adaptation. To compare the adaptation performance with respect to the number of speakers, two models trained using VCTK and LibriTTS datasets are used for speaker adaptation. We used 10 speakers of VCTK dataset as novel speakers for fine-tuning.

| Method | Pre-training | Fine-tuning | Transcript | PER | WER | EER | MCD | RMSE$_{f0}$ |
|---|---|---|---|---|---|---|---|---|
| GT | - | - | - | 4.26 | 16.69 | 4.14 | - | - |
| HierSpeech | VCTK | ✗ | - | 4.01 | 16.85 | 14.61 | 4.27 | 30.66 |
| HierSpeech | VCTK | ✓(1) | ✓ | 3.32 | 15.04 | 10.80 | 4.32 | 29.66 |
| HierSpeech | VCTK | ✓(5) | ✓ | 3.89 | 17.04 | 8.84 | 4.08 | 29.39 |
| HierSpeech | VCTK | ✓(10) | ✓ | 4.74 | 18.26 | 8.08 | 4.00 | 29.15 |
| HierSpeech | VCTK | ✓(20) | ✓ | 5.02 | 19.17 | 7.60 | 3.97 | 29.28 |
| HierSpeech-U | VCTK | ✓(1) | ✗ | 4.76 | 18.71 | 13.33 | 4.29 | 29.74 |
| HierSpeech-U | VCTK | ✓(5) | ✗ | 4.99 | 18.47 | 8.40 | 4.13 | 28.80 |
| HierSpeech-U | VCTK | ✓(10) | ✗ | 4.28 | 16.94 | 7.60 | 4.15 | 28.83 |
| HierSpeech-U | VCTK | ✓(20) | ✗ | 4.36 | 17.04 | 6.97 | 4.05 | 29.27 |
| HierSpeech | VCTK+LibriTTS | ✗ | - | 3.58 | 15.50 | 13.20 | 4.38 | 32.90 |
| HierSpeech | VCTK+LibriTTS | ✓(1) | ✓ | 3.44 | 15.91 | 16.80 | 4.47 | 29.98 |
| HierSpeech | VCTK+LibriTTS | ✓(5) | ✓ | 3.94 | 17.08 | 8.40 | 4.06 | 29.66 |
| HierSpeech | VCTK+LibriTTS | ✓(10) | ✓ | 4.95 | 18.32 | 7.20 | 4.03 | 30.02 |
| HierSpeech | VCTK+LibriTTS | ✓(20) | ✓ | 4.40 | 16.95 | 6.40 | 3.96 | 29.56 |
| HierSpeech-U | VCTK+LibriTTS | ✓(1) | ✗ | 4.16 | 16.99 | 12.00 | 4.31 | 29.61 |
| HierSpeech-U | VCTK+LibriTTS | ✓(5) | ✗ | 4.28 | 16.70 | 8.40 | 4.12 | 28.93 |
| HierSpeech-U | VCTK+LibriTTS | ✓(10) | ✗ | 4.45 | 17.15 | 7.60 | 4.12 | 30.02 |
| HierSpeech-U | VCTK+LibriTTS | ✓(20) | ✗ | 3.71 | 15.85 | 6.40 | 4.09 | 30.64 |

**Untranscribed text-to-speech**  We describe the results of the objective evaluation for speaker adaptation in Table 11. We compare the adaptation quality with different numbers of samples (1, 5, 10, and 20) and different numbers of pre-trained speakers (98 from VCTK and 1,151 from LibriTTS). Table 11 shows that the adaptation quality is improved with an increase in the number of samples.

**Phoneme predictor**  We conduct the ablation study of phoneme predictor. At first, we attempt to extract the linguistic representations without conditioning speaker information to extract the speaker-independent linguistic representations. However, adding speaker condition in linguistic encoder improves the model performance, indicating that a speaking style with the exception of voice is also trained from the speaker condition. Also, although the speaker conditioning is used in the linguistic encoder, the linguistic representations $z_l$ contains a little speaker information as shown in Table 2 and Figure 6. Following (Kim et al., 2021), we remove a bias parameter of phoneme predictor, which causes unstable training during mixed precision training. For better generalization, we attempt to train the phoneme predictor with label smoothing. However, it decreases the model performance, as it disturbs to optimize the KL divergence.

**Data augmentation**  For effective speech disentanglement, we use the data augmentation for the input of the linguistic encoder. We use the information perturbation of (Choi et al., 2021) to the waveform for the input of XLS-R. After data augmentation, the representations from the XLS-R may lose their speaker and pitch information, and then the linguistic representations are extracted from the linguistic-related information of XLS-R. However, this information perturbation decreases the model performance as shown in Table 12b. Also, our model already extracts the linguistic representations guided by phoneme predictor. Hence, the data augmentation for speech disentanglement is not necessary in our method.

Table 12: The results of ablation study and data augmentation.

(a) Ablation study for phoneme predictor.

| Model | CER | PER | WER |
|---|---|---|---|
| GT | 6.26 | 5.64 | 18.94 |
| HierSpeech | 6.77 | 6.25 | 20.89 |
| w.o speaker condition | 7.38 | 6.77 | 21.82 |
| w bias | 7.51 | 6.93 | 22.31 |
| w Label Smoothing | 7.99 | 7.60 | 23.00 |

(b) Experiment for data augmentation.

| Model | Augmentation | ratio | CER | PER | WER | EER |
|---|---|---|---|---|---|---|
| GT | | | 6.26 | 5.64 | 18.94 | 4.03 |
| HierSpeech | ✗ | - | 6.77 | 6.25 | 20.89 | 3.45 |
| HierSpeech | ✓ | 0.1 | 6.98 | 6.54 | 20.96 | 3.81 |
| HierSpeech | ✓ | 0.2 | 7.63 | 7.14 | 22.35 | 3.71 |
| HierSpeech | ✓ | 0.3 | 7.12 | 6.59 | 21.80 | 3.81 |
| HierSpeech | ✓ | 0.4 | 7.63 | 7.16 | 22.25 | 4.09 |
| HierSpeech | ✓ | 0.5 | 7.57 | 6.98 | 22.26 | 3.96 |
| HierSpeech | ✓ | 0.6 | 7.69 | 7.26 | 22.90 | 3.59 |
| HierSpeech | ✓ | 0.8 | 7.88 | 7.31 | 22.81 | 3.96 |
| HierSpeech | ✓ | 1.0 | 7.87 | 7.37 | 22.77 | 3.74 |

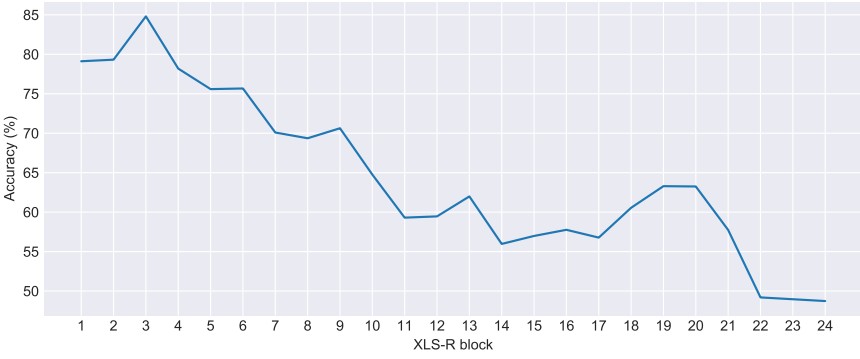

Figure 7: Speaker classification on self-supervised representations from different layers of XLS-R.

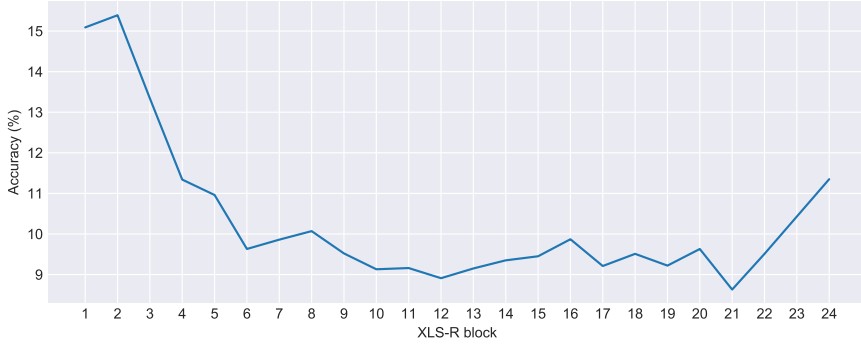

Figure 8: Speaker classification on linguistic representations from different layers of XLS-R.

**Analysis of the self-supervised speech representations for TTS**    Self-supervised speech representations contain rich information which is trained with large-scale speech dataset. Currently, wav2vec 2.0 (Baevski et al., 2020) is the most widely used model where the public can be easy to accessible to it, and it has shown the improved performance on downstream tasks such as ASR and speech translation. In this paper, we investigate these self-supervised representations to distill this rich information for text-to-speech task. Previous works show that each representation from the different layer of these model has different characteristics, and especially the representations from the middle layer contains the pronunciation information of speech (Shah et al., 2021). Our goal is to bridge the gap between text and speech through the additional representations, and the linguistic information such as pronunciation of speech may be appropriate to improve the pronunciation of synthesized speech. To demonstrate it, we train the model with each layer from the 24-layer transformer networks respectively. Table 13 shows that all layer with the exception of the 23th layer improve the model in ASR evaluation, indicating each representation contains rich information which is trained with large-scale speech dataset. Note that we fail to train the model with the representations from the 23th layer of XLS-R. For the hierarchical training of different information and the untranscribed text-to-speech, we should disentangle the linguistic and acoustic representations from speech. Hence, we conduct the speaker classification on each representation from the layer of XLS-R, and on each linguistic representation $z_l$ from each layer of XLS-R to measure the speaker information on the representations. We train the frame-level speaker classifier which consists of 4 Conv1D with kernel size of 7 followed by the projection layer. Figure 8 shows that the speaker classification results have an tendency to be decreased from the first layer to latter layer. However, the linguistic representations from the 12th layer of XLS-R have the lowest speaker classification accuracy in middle layer, indicating these representations may contain much more linguistic information which is trained with phoneme classification. In this regard, we use the representations from the 12th layer of XLS-R to extract the linguistic representations.

Table 13: Hyperparameter search for XLS-R (wav2vec 2.0) layer.

| Model | w2v layer | CER | PER | WER | EER |
|---|---|---|---|---|---|
| GT | | 6.26 | 5.64 | 18.94 | 4.03 |
| VITS | - | 12.53 | 12.24 | 30.62 | 3.85 |
| HierSpeech | 1 | 6.94 | 6.49 | 21.54 | 3.74 |
| HierSpeech | 2 | 7.01 | 6.65 | 21.39 | 3.74 |
| HierSpeech | 3 | 6.78 | 6.35 | 20.87 | 3.86 |
| HierSpeech | 4 | 6.93 | 6.44 | 21.05 | 3.96 |
| HierSpeech | 5 | 7.14 | 6.75 | 21.33 | 4.00 |
| HierSpeech | 6 | 7.00 | 6.55 | 21.16 | 3.86 |
| HierSpeech | 7 | 6.81 | 6.30 | 20.68 | 3.74 |
| HierSpeech | 8 | 6.87 | 6.32 | 20.89 | 3.56 |
| HierSpeech | 9 | 6.76 | 6.11 | 20.86 | 3.82 |
| HierSpeech | 10 | 6.72 | 6.12 | 20.71 | 3.59 |
| HierSpeech | 11 | 7.04 | 6.51 | 21.32 | 3.66 |
| HierSpeech | 12 | 6.77 | 6.25 | 20.89 | 3.48 |
| HierSpeech | 13 | 6.92 | 6.40 | 21.14 | 3.79 |
| HierSpeech | 14 | 6.89 | 6.36 | 20.60 | 4.01 |
| HierSpeech | 15 | 6.97 | 6.54 | 21.28 | 3.51 |
| HierSpeech | 16 | 6.73 | 6.35 | 20.81 | 3.62 |
| HierSpeech | 17 | 6.88 | 6.23 | 20.70 | 3.71 |
| HierSpeech | 18 | 7.13 | 6.62 | 21.41 | 3.74 |
| HierSpeech | 19 | 6.83 | 6.38 | 21.09 | 3.55 |
| HierSpeech | 20 | 7.53 | 7.09 | 22.00 | 3.96 |
| HierSpeech | 21 | 7.44 | 7.00 | 22.30 | 3.40 |
| HierSpeech | 22 | 7.62 | 7.18 | 22.34 | 4.03 |
| HierSpeech | 23 | - | - | - | - |
| HierSpeech | 24 | 7.42 | 6.85 | 21.68 | 3.75 |

# D    Baseline Models

**TTS**    We use an open source implementation of Tacotron 2[6], an official implementation of Glow-TTS[7], PortaSpeech[8], and VITS[9]. Since baseline models synthesize Mel-spectrogram unlike VITS and HierSpeech, we transform audio into Mel-spectrogram following (Kong et al., 2020) with a window size of 1024, hop size of 256, 1024 points of Fourier transform, and 22,050 Hz. For Tacotron 2, we use 32-dimensional speaker embedding which is concatenated with the output of text encoder following Skerry-Ryan et al. (2018). We train Tacotron 2 with batch size of 256 for 100k steps. For Glow-TTS, we condition 256-dimensional speaker embedding into the affine coupling layer in decoder and duration predictor to predict the speaker-specific duration. We train Glow-TTS with batch size of 128 for 960k steps. For PortaSpeech, we add 256-dimensional speaker embedding with the output of encoder and duration predictor. We train PortaSpeech with batch size of 64 with 320k steps. To convert Mel-spectrogram to waveform audio, we use the official implementation of pre-trained HiFi-GAN V1[10]. For VITS, we use the same speaker conditioning of ours and train the model with batch size of 256 with 600k steps. For VITS and HierSpeech, we sample 32 sequences from the whole $z_a$, and upsample it by 256x which is the same size as hop size for STFT. We use VCTK and *train-clean-360* and *train-clean-100* subset of LibriTTS dataset. Both dataset are licensed under the Creative Commons Attribution 4.0.

**VC**    We use the official implementation of AutoVC[11] and VoiceMixer[12]. Both models are trained with Mel-spectrogram segments of 192 frames during training. We train the AutoVC with the information bottleneck of 32 frames and a batch size of 16 for 100k steps. For VoiceMixer, we train the model with a batch size of 64 for 150k steps. We also use the pre-trained HiFi-GAN V1 to convert the Mel-spectrogram to waveform audio.

---

[6]https://github.com/NVIDIA/tacotron2

[7]https://github.com/jaywalnut310/glow-tts

[8]https://github.com/NATSpeech/NATSpeech

[9]https://github.com/jaywalnut310/vits

[10]https://github.com/jik876/hifi-gan

[11]https://github.com/auspicious3000/autovc

[12]https://github.com/anonymous-speech/voicemixer/tree/main/code

# E Evaluation Details

**Mean opinion score**  We include the details of instructions of MOS evaluation in Figure 9.

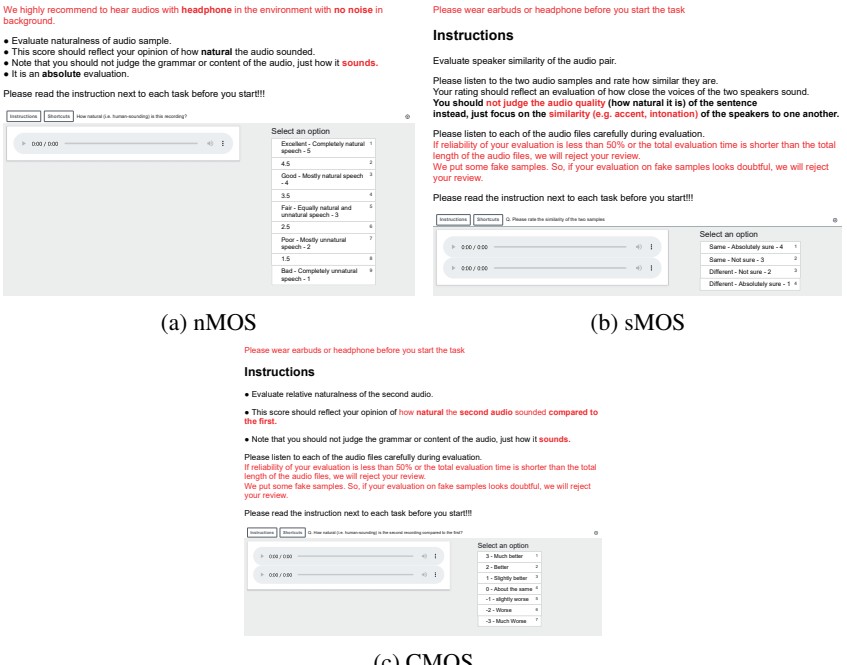

(a) nMOS

(b) sMOS

(c) CMOS

Figure 9: The screenshots of MOS evaluation. $0.1 per 1 hit is paid to participants for nMOS (7 samples) and $0.01 per 1 hit is paid to participants for sMOS and CMOS.

**Automatic speech recognition**  We use the pre-trained wav2vec 2.0 model (base model[13]) to evaluate the character error rate, phoneme error rate, and word error rate (WER). We calculate the WER based on the character prediction results to evaluate the pronunciation without language model.

**Automatic speaker verification**  We use the pre-trained ASV model[14] to evaluate the speaker similarity. The Fast ResNet-34 model (Chung et al., 2020) is used to extract the features, and the similarities of features are compared for the verification results. The model was trained by VoxCeleb2 (Chung et al., 2018) with online data augmentation (Heo et al., 2020). The online data augmentation improves the model performance from EER of 2.1792 to 1.1771 as described in (Heo et al., 2020).

**Mel cepstral distortion**  We extract the Mel-frequency cepstral coefficients (MFCCs) by discrete cosine transform to raw waveform[15]. Then, we calculate the MCD between ground-truth and synthesized speech from the first 13 MFCCs by using dynamic time warping (DTW) to align the different length of sequences.

**F0 root mean square error**  We extract the fundamental frequency F0 by World vocoder[16]. Then, we compute the $l2$ distance between the ground-truth and synthesized speech for $\text{RMSE}_{f0}$. We use the DTW to align two sequences.

**Average differences of the utterance duration**  We conduct DDUR evaluation presented in (Zhang et al., 2019) to evaluate the duration prediction performance. For fair evaluation, we trim the silence of audio, and then we calculate the average absolute differences of duration between ground-truth and synthesized speech.

---

[13] https://huggingface.co/docs/transformers/model_doc/wav2vec2

[14] https://github.com/clovaai/voxceleb_trainer

[15] https://github.com/MTG/essentia/

[16] https://github.com/JeremyCCHsu/Python-Wrapper-for-World-Vocoder