# OpenReview forum: "HierSpeech: Bridging the Gap between Text and Speech by Hierarchical Variational Inference using Self-supervised Representations for Speech Synthesis"
_NeurIPS.cc/2022/Conference — NeurIPS 2022 Accept_

### Official Review · Reviewer_oiLb · 2022-07-08

**Rating:** 7
**Confidence:** 4
**Soundness:** 4 excellent
**Presentation:** 4 excellent
**Contribution:** 3 good

**Summary:**

This paper studies end-to-end text-to-speech synthesis and few-shot speaker adaptation using untranscribed speech. In particular, the authors extend from the VITS model and leverage self-supervised speech representations as the intermediate latent variable to bridge the gap between text and acoustic features. An additional phoneme prediction loss is used to refine self-supervised representation, making it closer to flow-transformed acoustic latent z_a. In addition, the injection of the intermediate latent inferred from self-supervised model enables speaker adaptation using untranscribed data.

**Questions:**

1. Adding PP to the VITS posterior encoder is helpful. The posterior encoder in VITS is analogous to the acoustic encoder in HierSpeech. Have the authors tried adding PP loss to z_a also?
2. What is “sliced” z_a?
3. What is the “global conditioning” in Section 2.3 and how is it used? Is it different from speaker embedding? If it is different, does training on transcribed data require it?
4. What is the speaker encoder and how is it trained?


**Limitations:**

Yes, the authors discussed it. It is nicely addressed and mitigation strategies are presented.

**Strengths And Weaknesses:**

Strengths
1. An interesting approach to integrate linguistic representations learned by SSL models into the end-to-end TTS pipeline
2. Provides a neat way of speaker adaptation with untranscribed data
3. Phoneme prediction loss seems to help not only HierSpeech but also VITS
4. Good ablation studies comparing different layers of wav2vec representations for HierSpeech and favorable results compared to the baselines (VITS).

Weaknesses
1. Some details are missing. See the question section.

---

> ### Author Response · Authors · 2022-07-31
> **Responses to Reviewer oiLb**
>
> We appreciate for your helpful comments and suggestions. We have provided responses to your questions below to address your concerns.
>
> >Q1. Adding PP to the VITS posterior encoder is helpful. The posterior encoder in VITS is analogous to the acoustic encoder in HierSpeech. Have the authors tried adding PP loss to z_a also?
>
> As you mentioned, adding the phoneme predictor to the acoustic encoder is helpful because it improves the capability of linguistic information in latent representations. Table 7 also shows that VITS trained with phoneme predictor has better performance in phoneme error rate (PER) and word error rate (WER). However, in our experimental settings, we found that adding the phoneme predictor to the acoustic encoder directly decreases the audio quality with more noise. To analyze this phenomenon, we evaluate the reconstruction quality of each model in terms of Mel-spectrogram reconstruction error (Mel distance) and perceptual evaluation of speech quality (PESQ). To compare the quality of reconstructed audio, we used the same models which we used for ablation studies (trained during 300k steps). For PESQ, we compared each model in both wide and narrow bands. For Mel distance, we calculated the l1 distance between ground-truth Mel-spectrogram from target audio and reconstructed Mel-spectrogram from the reconstructed audio.
>
> (Reconstruction flow: target audio --> linear spectrogram --> Acoustic encoder --> z_a --> Decoder --> reconstructed audio)
>
> |Model|Wide Band PESQ (↑)|Narrow Band PESQ (↑)|Mel distance (↓)|
> |------|:---:|:---:|:---:|
> |VITS|2.03|2.57|0.46|
> |VITS+Phoneme Predictor|2.00|2.55|0.49|
> |HierSpeech|2.12|2.67|0.44|
>
> The results show that although adding a phoneme predictor to the acoustic encoder improves the linguistic capability in acoustic representation, it also degrades the audio quality. Hence, we did not add the phoneme predictor to the acoustic encoder. Compared with ground-truth audio, our model has almost similar phoneme error rate and word error rate, so we think that it is not necessary to use an additional phoneme predictor in our model.
>
> > Q2, What is “sliced” z_a?
>
> Due to the computational complexity, most end-to-end models [1, 2, 3] use the windowed generator training by upsampling the partial sequences in the decoder. In our case, we sample 32 sequences from the whole z_a, and upsample it by 256x which is the same size as hop size for STFT. We will add the details about the sliced z_a in the revised paper.
>
> > Q3. What is the “global conditioning” in Section 2.3 and how is it used? Is it different from speaker embedding? If it is different, does training on transcribed data require it?
> Q4.What is the speaker encoder and how is it trained?
>
> As you mentioned, it is the same as speaker embedding. For the multi-speaker settings, we add the global speaker embedding to the residual block of the acoustic/linguistic encoder, residual block in normalizing flow, stochastic duration predictor, and the input of the decoder. For zero-shot or few-shot scenarios, we use the speaker (style) encoder to extract the speaker embedding from ground-truth speech. The speaker encoder is trained with the entire model jointly. We use the linear spectrogram as an input of the speaker encoder, and the speaker embedding with 256 dimensions is extracted. We will add the details of the speaker encoder and we will replace “style encoder” with “speaker encoder”. Specifically, the speaker encoder consists of two fully-connected layers with 256 hidden units, two one-dimensional convolutional networks with a residual connection (filter size of 256 and kernel size of 5), a multi-head self-attention module, and a projection layer followed by temporal average pooling.
>
> >Reference
>
> [1] Jeff Donahue, Sander Dieleman, Mikołaj Bińkowski, Erich Elsen, and Karen Simonyan, “End-to-End Adversarial Text-to-Speech,” International Conference on Learning Representations (ICLR), 2021.
>
> [2] Yi Ren, Chenxu Hu, Xu Tan, Tao Qin, Sheng Zhao, Zhou Zhao, and Tie-Yan Liu, “FastSpeech 2: Fast and High-Quality End-to-End Text to Speech,” International Conference on Learning Representations (ICLR), 2021.
>
> [3] Jaehyeon Kim, Jungil Kong, and Juhee Son, “Conditional Variational Autoencoder with Adversarial Learning for End-to-End Text-to-Speech,” International Conference on Machine Learning (ICML), 2021

---

> > ### Comment · Reviewer_oiLb · 2022-08-09
> > **Thank you for the response**
> >
> > I have read the authors' responses which addressed my questions. I will keep the original rating.

---

### Official Review · Reviewer_Pops · 2022-07-09

**Rating:** 6
**Confidence:** 3
**Soundness:** 3 good
**Presentation:** 3 good
**Contribution:** 3 good

**Summary:**

In order to address the challenge in TTS that it is hard to infer both the linguistic and acoustic attributes from the text directly, the paper proposes HierSpeech, which is a high-quality end-to-end TTS system based on a hierarchical conditional variational autoencoder (VAE) utilizing self-supervised speech representations. The pipeline firstly extracts self-supervised linguistic representation from text, and then converts it to acoustic representations, and finally generates waveform.

**Questions:**

I think the author can discuss the difference of propose method with method in [1], which also makes use of self-supervised linguistic representation. Whether the only difference in the pipeline is that the method in [1] leverages a spectrum synthesizer and hifi-gan but the proposed method incorporates hifi-gin into HierSpeech pipeline (since the HierSpeech also utilizes GAN loss for waveform synthesis)?


[1] VQTTS: High-Fidelity Text-to-Speech Synthesis with Self-Supervised VQ Acoustic Feature


**Strengths And Weaknesses:**

Strengths:
1. The paper introduces self-supervised linguistic representation from wav2vec to help the TTS.
2. The experiments verify the effectiveness of proposed pipeline. The paper also discusses the potential of HierSpeech on few-shot TTS and voice conversion.
3. The paper is well-written and easy to understand.

Weaknesses:
There is no obvious weakness.

---

> ### Author Response · Authors · 2022-07-31
> **Responses to Reviewer Pops**
>
> We appreciate for your helpful comments and suggestions. We have provided responses to your questions below to address your concerns.
>
> >Q1. I think the author can discuss the difference of propose method with method in [1], which also makes use of self-supervised linguistic representation. Whether the only difference in the pipeline is that the method in [1] leverages a spectrum synthesizer and hifi-gan but the proposed method incorporates hifi-gin into HierSpeech pipeline (since the HierSpeech also utilizes GAN loss for waveform synthesis)?
>
> Thanks for your suggestion. We will add VQTTS paper [1] in related works, which was accepted in Interspeech 2022 after our submission. We agree that this paper also utilizes the self-supervised speech representation for speech synthesis, however, we can discuss the differences in our method as below
>
> 1. As you mentioned, while VQTTS is a two-stage text-to-speech system, we successfully integrate self-supervised speech representation into the singe-stage end-to-end text-to-speech pipeline.
>
> 2. VQTTS replaces Mel-spectrogram with self-supervised vector quantized representation for the acoustic feature. However, as shown in our experiments, self-supervised speech representation has lack of acoustic capability. Hence, when using these representations for acoustic representation, it results in mispronunciation and over-smoothing problems in the synthesized speech, and this occurs more frequently in the multi-speaker scenarios. In our ablation study, Table 7 shows that using self-supervised representation directly causes performance degradation. This also means that the spectrogram contains more acoustic information than self-supervised representation. In the paper [1], they also mentioned the limitation in terms of reconstruction performance due to the information loss brought by quantization which results in a much lower PESQ score than that of using Mel-spectrogram. Because we had the same problem in early experiments, we designed our model by hierarchical structure using both linguistic representation from self-supervised representation and acoustic representation from spectrogram, which is important for expressive speech synthesis. In my opinion, to alleviate the problems mentioned above, VQTTS has additional networks using additional CNNs and Conformer blocks, and they also used additional prosody features from text sequences. So, we are not entirely sure that the performance of VQTTS is improved by solely using self-supervised representation as acoustic representation. We see that our hierarchical structure can be applied to VQTTS to improve the expressiveness in their synthesized speech.
>
> 3. Unlike VQTTS, we utilize the self-supervised speech representation to disentangle linguistic information and acoustic information, and to learn each attribute hierarchically. With the hierarchical structure, we achieve better performance in the linguistic and acoustic metrics. Also, our model can extract the linguistic representation from audio-only data without text transcripts by demonstrating the untranscribed speaker adaptation. But VQTTS still needs the text-audio paired dataset.
>
> We will add this discussion in the revised paper.
>
> >Reference
>
> [1] Chenpeng Du, Yiwei Guo, Xie Chen, and Kai Yu, "VQTTS: High-Fidelity Text-to-Speech Synthesis with Self-Supervised VQ Acoustic Feature," Interspeech, 2022.

---

### Official Review · Reviewer_SZqL · 2022-07-15

**Rating:** 6
**Confidence:** 3
**Soundness:** 2 fair
**Presentation:** 2 fair
**Contribution:** 3 good

**Summary:**

The authors propose a novel speech synthesis model which utilizes untranscribed speech data to improve the representation of the linguistic information. The authors combine the representation of wave2vec (self-supervision) as an input to a linguistic encoder which is used as a prior for an acoustic VAE. The authors augments the VAE usual ELBO with multiple terms to improve the audio quality, and also add KL divergence term between the acoustic prior and the linguistic prior (which is conditioned on text) to support speech synthesis.
The authors perform multiple experiments to demonstrate the effectiveness of their method, and ablation studies to justify their design choices. The authors also demonstrate how the proposed method allows untranscribed text-to-speech by fine tuning the acoustic encoder over audio-only data.

**Questions:**

* Lines 85-92 :This paragraph repeats the previous paragraph?

* Line 121: NVAE is not a general hierarchical VAE but rather a particular hierarchical VAE for images

* Line 127: Eq. 2 does not make sense. As it currently stands, the rightmost term has no effect on the generative process or the acoustic posterior.
In practice, this effect is established via the weight sharing of p_theta_a(z_a|x_w2v) = q_phi_l(z_lx_w2v) but this critical information is rather easy to miss. Please fix it by adapting equation 2 to explicitly show the effect of the weight sharing.

I add here an example of formulating Eq. 2 with explicit weight -sharing: replace q_phi_l(z_l|w_w2v) with  p_theta_a(z_a|x_w2v) and separate Eq. (2) into two terms, ELBO + and KL( p(z_a|x_wv2) || p(z_a|c) )
This formulation explicitly shows that the conditional prior p(z_a|x_w2v) is regularized to match the linguistic conditional prior p(z_l |c)

* Line 128: It could be useful to define A here

* Line 283: extra "The"

* Line 306: Which KL divergence problem did you encounter? Posterior collapse? Other problems?

**Limitations:**

The authors discussed the limitations of the proposed method.

**Strengths And Weaknesses:**

Strengths:
* The audio quality of the provided samples is very good compared to competing methods.
* The proposed model demonstrates consistent improvement over existing methods.
* The proposed model supports untranscribed text-to-speech.

Weaknesses:
* The formulation of the acoustic VAE is misleading, since the "weights sharing" between the acoustic prior and the linguistic posterior is critical, but is not part of Eq. 2 (the ELBO)
* The system overview lacks clarity, and adding all the loss terms (instead of a single KL term)  could help.

---

> ### Author Response · Authors · 2022-07-31
> **Responses to Reviewer SZqL**
>
> We appreciate your helpful comments and suggestions. We have provided responses to your questions below to address your concerns.
>
> >Q1. The formulation of the acoustic VAE is misleading, since the "weights sharing" between the acoustic prior and the linguistic posterior is critical, but is not part of Eq. 2 (the ELBO)
>
> Thanks for your suggestion about Equation 2. We acknowledge that this part may be misleading since we do not emphasize enough the weight-sharing between the acoustic prior and the linguistic posterior. As shown in Figure 2, the linguistic encoder is used to extract acoustic prior, which is also used as linguistic posterior. Although we also stated it in Line 128 and 152-153, we agree it is easy to miss as you mentioned. Following your suggestion, we will clarify this equation in the final manuscript. To avoid confusion, we will also emphasize the weight-sharing part in the revised paper.
>
> >Q2. The system overview lacks clarity, and adding all the loss terms (instead of a single KL term) could help.
>
> Thanks for your suggestion. We will redraw the system overview in Figure 2 by adding all the loss terms.
>
> >Q3. Lines 85-92 :This paragraph repeats the previous paragraph?
>
> Thank you for your advice. We will remove the same content which is stated in the previous paragraph, and revise it in the final manuscript.
>
> >Q4. Line 121: NVAE is not a general hierarchical VAE but rather a particular hierarchical VAE for images
>
> Thanks for your advice. We acknowledge that NVAE is a particular hierarchical VAE for images, not a general hierarchical VAE. In the recent speech domain, BVAE-TTS [1] and PVAE-TTS [2] successfully adopted NVAE to the text-to-speech system by learning hierarchical latent representations to increase expressiveness, so we overstated it as general hierarchical VAE. To avoid confusion, we will clarify this term by removing “general”.
>
> >Q5. Line 128: It could be useful to define A here
>
> Thank you for suggestion. We will add the definition of  A here in the final manuscript.
>
> >Q6. Line 283: extra "The"
>
> Thanks for your comments. We will remove the extra “The” in the revised version.
>
> >Q7. Line 306: Which KL divergence problem did you encounter? Posterior collapse? Other problems?
>
> When we trained the model with label smoothing and data augmentation, we experienced that the KL divergences increases more than that of the model without using them. It was not the case of posterior collapse, but it resulted in the audio quality degradation of synthetic speech. We will clarify it in the revised paper.
>
> To avoid another KL divergence problem, we found that it is important to remove the silences of audio data for each particular dataset. Specifically, the audio of VCTK dataset has leading and trailing silences. Without removing silences, we had trouble training the model because the KL divergence between the linguistic posterior and prior increases, and therefore the model cannot synthesize speech from text. After we trim the audio under the threshold to determine silences for VCTK dataset, the model is successfully trained. Although these details are included in our attached source code, we will add these details in the appendix for better reproducibility. We recommend using 20dB as a threshold for VCTK and LibriTTS dataset, and not removing the silence for LJSpeech dataset.
>
> >Reference
>
> [1] Yoonhyung Lee, Joongbo Shin, and Kyomin Jung, "Bidirectional variational inference for non-autoregressive text-to-speech," International Conference on Learning Representations (ICLR), 2020.
>
> [2] Ji-Hyun Lee, Sang-Hoon Lee, Ji-Hoon Kim, and Seong-Whan Lee, "PVAE-TTS: Adaptive Text-to-Speech via Progressive Style Adaptation," ICASSP 2022-2022 IEEE International Conference on Acoustics, Speech and Signal Processing (ICASSP), 2022.

---

### Meta-Review · Area_Chair_CDBx · 2022-09-09

**Recommendation:** Accept
**Confidence:** Certain

**Metareview:**

all reviewers agree
* the paper is interesting and novel
* the proposed method has solid experiments and good results
* paper is well written

this paper should be accepted to the conference.

**Award:**

No

---

### Decision · Program_Chairs · 2022-09-14

Accept